# Prevalence and associated factors of postpartum anemia after cesarean delivery in public hospitals of Awi zone, North West Ethiopia, 2023; a cross-sectional study

**Gebretsadek Habtamu[1], Asmare Talie[1], Tinsae Kassa[1], Dawit Misganaw Belay** [2]*

1 Department of Midwifery, College of Health Sciences, Debre Markos University, Debre Markos, Ethiopia,
2 Department of Midwifery, College of Health Sciences, Assosa University, Assosa, Ethiopia

* davemisganaw.dm@gmail.com

**Data Availability Statement:** All data are in the manuscript and/or supporting information files.

## Abstract

### Background

Anemia is a serious global public health problem, especially in developing nations. Anemia during pregnancy is appropriately recognized, whereas postpartum anemia especially after cesarean delivery in Ethiopia has received very little attention. Due to this it leads to poor quality of life, palpitations, an increase in maternal infections, exhaustion, diminished cognitive function and postpartum depression. Therefore, this study aimed to assess the prevalence and associated factors of postpartum anemia after cesarean delivery in public hospitals of Awi zone, North West Ethiopia, 2023.

### Method

A hospital-based cross-sectional study was conducted among 395 mothers who gave birth by cesarean delivery from May 1–30, 2023. Data were collected using a pretested checklist. A simple random sampling technique was used to select study participants. Then the data were entered into EPI-data version 4.6 and exported to the SPSS version 25 for analysis. A logistic regression model was fitted to assess the association between outcome and explanatory variables. Variables with a p-value of 0.25 or less in bivariable analysis were candidates for multivariable analysis and P-value < 0.05 in multivariable analysis was considered to declare a result as statistically significant in this study.

### Result

The prevalence of postpartum anemia after cesarean delivery was 18.9% (95% CI (15.1, 23.1)) with a response rate of 97.97%. Being primipara (AOR = 0.47,95%CI = 0.24,0.92), indication for current C/S (malpresentation) (AOR = 0.29,95%CI = 0.09,0.90), having pre-operation hemoglobin level <11g/dl (AOR = 14.5;95% CI = 4.11,51.16) and having medical complication during current pregnancy (AOR = 5.95,95%CI = 1.88,18.83) were significantly associated with postpartum anemia after cesarean delivery.

**Funding:** The author(s) received no specific funding for this work.

**Competing interests:** The authors have declared that no competing interests exist.

**Abbreviations:** APH, Antepartum Hemorrhage; ANC, Antenatal Care; AOR, Adjusted Odds Ratio; CD, Cesarean Deliveries; CS, Cesarean Section; Hgb, Hemoglobin; IFA, Iron and Folic Acid; PPA, Postpartum Anemia; PPH, Postpartum Hemorrhage; SPSS, Statistical Product Service and Solution.

## Conclusion

The findings of the study show that the prevalence of postpartum anemia after cesarean delivery is a mild public health problem. Therefore, promoting the benefits of early detection and management of pregnancy complications such as predelivery anemia and medical complications is crucial.

## 1. Introduction

The World Health Organization (WHO) defines anemia as a condition in which the number of red blood cells, or the concentration of hemoglobin within red blood cells, is lower than normal [1]. Although there is no universally consensual definition of postpartum anemia (PPA), it can be deduced from the definitions offered by various scholars, depending on the duration of the postpartum period. It can be defined as Hgb < 10 g/dl, Hgb < 11g/dl, and Hgb < 12g/dl cut-off values within the first 48 hours of delivery, at 1 week and 6 weeks of postpartum duration, respectively [1–3].

Postpartum anemia is usually brought on by both chronic iron deficiency that has existed throughout pregnancy and bleeding during childbirth. In the third trimester of pregnancy, due to increased nutrient expenditure on the baby's growth causes iron deficiency [2,4]. On the other hand, the woman's body mass and her total blood volume affect the amount of blood loss. It also depends on any further medical issues she might have. For instance, a woman with a cardiac problem experiences greater decompensation with less blood loss [5].

Anemia is a serious global public health problem that particularly affects pregnant and postpartum women [6]. Anemia accounts for 7% of maternal mortality due to indirect causes and 2.3% of all causes, with indirect causes accounting for 35% of all causes of maternal deaths globally [6,7]. Even though anemia is a global problem, there are regional variations. Western Sub-Saharan Africa, South Asia, and Central Sub-Saharan Africa regions had the highest burdens [8]. More specifically, the proportion of postpartum mothers who have anemia ranges from 10% to 30% in developed countries and from 50% to 80% in developing countries [1,9]. In East African nations, about 36.5% of postpartum women were especially vulnerable to postpartum anemia (PPA) [10]. The prevalence of PPA in Ethiopia also ranges from 11.6% in Addis Ababa to 58.7% in the Somali Region [11].

Women who undergo a cesarean section may be more vulnerable to postpartum anemia because they have a higher risk of postpartum hemorrhage (PPH) than women who give birth vaginally [12,13].

Evidence shows that cesarean section increases postpartum hemorrhage largely through increased risk of uterine atony and minimally through severed vessels while opening the abdominal cavity [14–16]. Postpartum anemia is more likely to occur in pregnant women who have anemia during pregnancy, especially in the third trimester, excessive intrapartum blood loss, younger women, and those who did not take an iron supplement while pregnant [17]. Up to half of the women who miss prenatal iron supplements develop anemia within 48 hours after delivery [18].

Postpartum anemia (PPA) is associated with poor quality of life, palpitations, an increase in maternal infections, exhaustion, diminished cognitive function, and postpartum depression. These outcomes may result in poor mother-child bonding, an inability to care for and breastfeed an infant, or slow baby development [1,19,20].

Several studies have been conducted on anemia during pregnancy [21–23]. However, these studies have provided limited information about the prevalence of postpartum anemia especially among women undergoing cesarean section, even when global trends show increasing cesarean section rates. In light of this, little is known regarding postpartum anemia among postpartum women in Ethiopia, particularly among women undergoing cesarean section, which has not been studied. So, this study might provide insight into postpartum anemia to healthcare providers to propose targeted screening and intervention measures for those whose hemoglobin level <11 gm/dl. The study will also provide baseline data for policymakers concerned governmental entities, nongovernmental organizations, and other concerned stakeholders to plan and act to prevent and minimize postpartum anemia after cesarean section. This study aimed to assess the prevalence and associated factors of postpartum anemia after Cesarean delivery in public hospitals of Awi zone, North West Ethiopia, 2023.

## 2. Methods and materials

### 2.1 Study area and period

This study was conducted in Awi Zone public hospitals. which is part of Amhara regional state northwest Ethiopia, which has eleven districts and three-town administrations. There are five hospitals in Awi Zone (Injibara general Hospital, Chagni primary Hospital, Dangila primary Hospital, Gimjabet primary Hospital and Jawe primary Hospital) and the Zone has a total population of above one million as population census conducted by central statistical agency of Ethiopia in 2007 and located in North-western part of Ethiopia and which is far 477 km from Addis Ababa a capital city of Ethiopia [24]. In 2021, there were 1995 deliveries, and in 2022, there were 2688 deliveries, with cesarean section (CS) being performed in five public hospitals of Awi zone. The Study was conducted from May 1 to May 30, 2023.

### 2.2 Study design and population

Hospital based cross-sectional study design was conducted. All mothers who gave birth by Cesarean delivery in public Hospitals of Awi Zone were our source population. All cesarean deliveries during the last two years (January 1, 2021 to December 31, 2022) in public Hospitals of Awi Zone were our study population. However, those who had pre-operative severe anemia and those who received blood transfusion were excluded.

### 2.3 Sampling methods

**2.3.1 Sample size calculation.** *2.3.1.1 Using single population proportion formula.* A single population proportion formula used to estimate the sample size and using the following assumptions: from previous study done in Debre Berhan, Ethiopia the proportion of postpartum anemia after Cesarean delivery was 18%. confidence interval of 95%($Z\alpha/2 = 1.96$) and 4% of marginal error (d = 0.04). So, the sample size becomes **355.**

So, with the above inputs, the maximum sample size we got for this study was 355, which is calculated from the single population proportion formula. Therefore, the final sample size becomes 395 study participants, including the non-response rate.

**2.3.2. Sampling procedure.** All the public hospitals in the Zone were included. The card numbers of all mothers who gave birth through cesarean section at Awi Zone Public Hospital for the last two years before the study (January 1, 2021, to December 31, 2022) were traced from the hospital's delivery log book registry and were listed. A two-year report of birth through cesarean section is collected from each hospital health management information system and summed up to calculate the proportion. Then the total sample size was proportionally

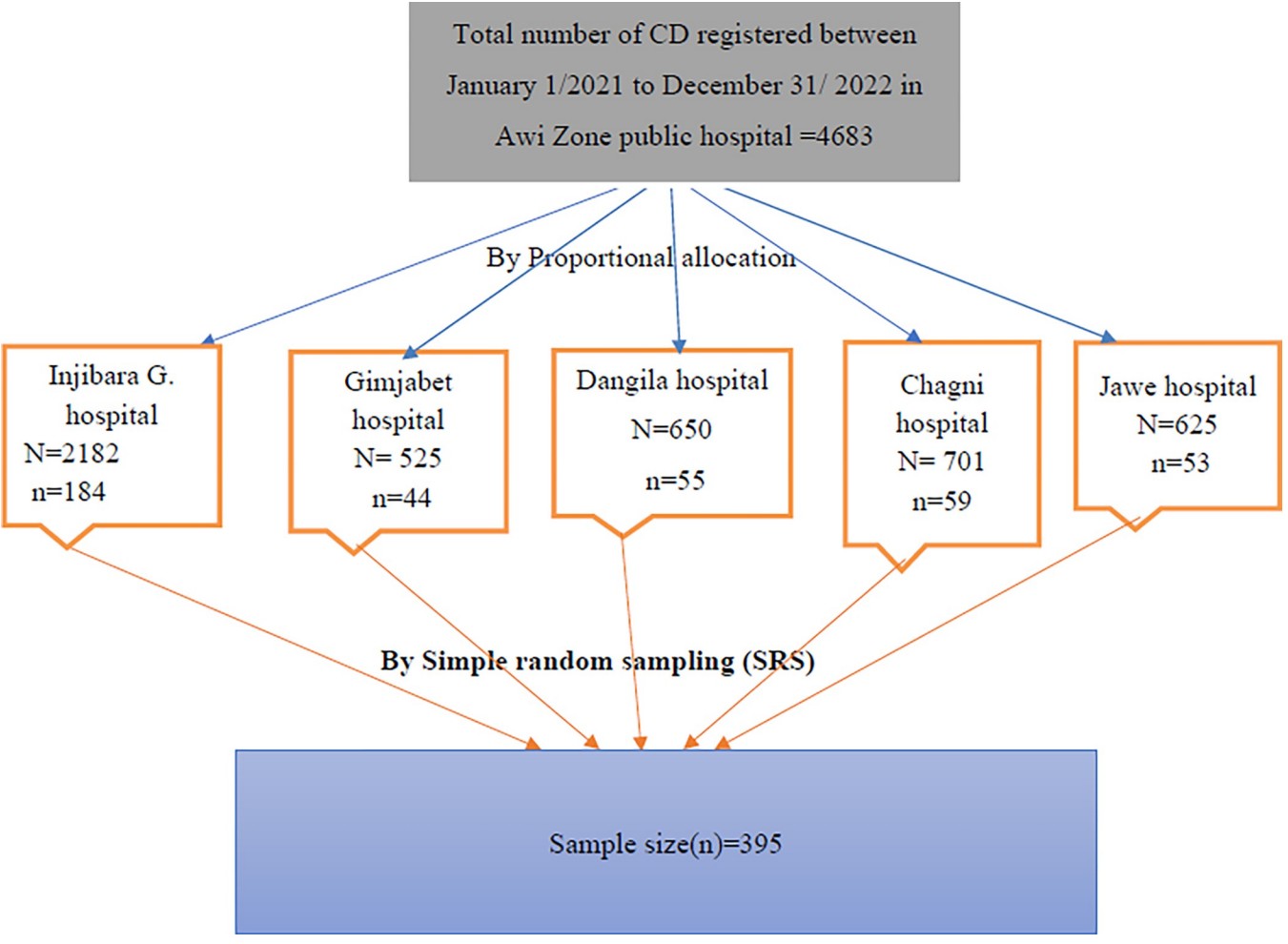

**Fig 1. Schematic presentation of sampling procedures for the selection of study subjects at Awi zone public hospitals, Ethiopia, 2023.**

allocated to each public hospital. Then, a computer-generating simple random sampling technique was used until the allocated sample for each facility was fulfilled (**Fig 1**).

## 2.4 Operational definition of variable

The extent of postpartum anemia after Cesarean delivery as a dependent variable, defined by World Health Organization (WHO) criteria as a postpartum Hgb level of less than 11 g/dL, measured closest to the day of hospital discharge [25].

Incomplete Card: mothers' card which was not contain major information about mother's condition (post CS hemoglobin level, pre-CS hemoglobin level).

Surgical site infection: An infection that happens within the first few days after surgery of abdominal skin and the underlying tissues [26].

Severe postpartum anemia; defined by World Health Organization (WHO) criteria as a postpartum Hb level of less than 8 g/dL, measured closest to the day of hospital discharge [27].

## 2.5 Study variables

### 2.5.1 Dependent variable.

➢ Prevalence of postpartum anemia after Cesarean delivery.

### 2.5.2 Independent variable.

➢ Socio- demographic related characteristics: age, residence

➢ Obstetric related characteristics: ANC follow up, gravidity, parity, Utilization of IFA, Type of Pregnancy, Number of previous C/S, C/S type, indication for current C/S, type of uterine incision, weight of newborn, APH in the current pregnancy, PPH in the current pregnancy, medical complication in current pregnancy.

## 2.6 Data collection procedure

A structured data collecting checklists was prepared according to the objectives of the study adapted from relevant literatures [28,29] in English language. By reviewing charts, necessary adjustment was made to fit the local condition. The main contents of the checklists were including: socio-demographic variables, and obstetrics related variables and medical conditions related variables. Two supervisors (BSc midwifes) and Five data collectors (diploma midwifes) were recruited.

## 2.7. Data quality control

To assure the quality of data, data collectors and the supervisors were trained for one day by the principal investigator on the study checklist, consent form and data collection procedure. Furthermore, quality of data was assured by pretesting checklist. A pretest was done on other health facility one week before the main data collection. Then the checklist was modified based on pretest finding. In addition, the completeness, accuracy and consistency of collected data will be checked on daily bases during the data collection time by supervisors and principal investigator. Supervisor and principal investigator were closely following the data collection process.

## 2.8 Data processing and analysis

After checking the data manually for completeness and consistency, the data were cleaned, coded and entered using Epi-data version 3.1 statistical Software. Then data were exported to SPSS version 25 statistical Software for analysis. Descriptive statistics were computed for variables using frequencies, percentages, mean and standard deviation. Graphical presentation such as bar graph, line graphs and pie charts were used to present the findings of the study. Both bivariable and multivariable logistic regression analysis were employed to determine association between the independent variables and the dependent variable. Bivariable logistic regression were done to identify relationship between one independent variable and outcome variable. Those variables with p-value of less than 0.25 during bivariable were fitted into multivariable logistic regression model to identify variables independently associate with outcome variable. Odd ratio with 95% confidence interval and p value were calculated. Variables having P-value < 0.05 in the multivariable logistic regression analysis were considered as associated factors for postpartum anemia after cesarean delivery. The final model's fitness was checked by conducting the Hosmer-Lemeshow Goodness of Fit test, and a multicollinearity test was performed to check the relationships between the independent variables.

## 2.9 Ethical consideration

Ethical approval was obtained from the Debre Markos University, College of Medicine and Health Science, Institutional Research Ethics Review Committee (IRERC) with Reference

**Table 1. Socio-demographic characteristics of respondents in public hospitals of Awi Zone, Ethiopia, 2023.**

| Variables | Category | Frequency | Percentage (%) |
|---|---|---|---|
| Age | 15–24 | 89 | 23 |
| | 25–34 | 244 | 63 |
| | ≥35 | 54 | 14 |
| Residence | Urban | 170 | 43.9 |
| | Rural | 217 | 56.1 |

number HSC/RCS/144/11/15). Also, A permission letter was secured from Amhara Regional Health Bureau and each hospital. Moreover, individual consent was not applicable since it is a retrospective study of medical records (record review). The ethics committee waived the requirement for informed consent. Finally, the confidentiality of the information and privacy of study participants was maintained.

# 3. Results

## 3.1 Socio -demographic characteristics of respondents

Among 395 total sample of mother after CS delivery, 387 were participated in this study with a response rate of 97.97%. The mean age of participants was 28.37(SD ± 5.11), ranging from 16 to 41 years (**See Table 1**).

## 3.2 Obstetrics related characteristics of mother

Among the total participants, mothers who had ANC follow up were 325(84%). Nearly three-fourth (72.9%) of them were multigravida and two hundred thirty-nine (61.8%) of them were multipara. The majority, 356 (92%) of the current pregnancy were singleton (**See Table 2**).

The most common type of CS was emergency 318 (82.2%) followed by elective type 69 (17.8%).

Among CS delivery, 24 (6.2%) women have one previous CS scar. The most common indications of cesarean delivery were fetal distress (Non-Reassuring Fetal Heart Rate (NRFHR)) 118(30.5%), followed by prolonged labor (abnormal labor) 85(22%), preeclampsia-eclampsia 53 (13.7%), malpresentation 50(12.9%), and previous CS scar 43 (11.1%) (**See Table 2**).

Regarding medical comorbidity during current pregnancy, among 387 mothers, 20 (5.2%) had pre-operation anemia (there hemoglobin levels were <11 gm/dL), whereas 367 (94.8%) of them had hemoglobin level of 11 gm/dL and above (**See Table 2**). Sixty-seven (17.3%) of mothers had medical complication in current pregnancy. Of these, pregnancy induced hypertension (PIH) was the most prevalent medical complication which accounts 51(56%) (**Fig 2**).

## 3.3 Prevalence of postpartum anemia after cesarean delivery

Among 387 postpartum mothers who gave birth through cesarean section, 18.9% (95% CI (15.1, 23.1)) were anemic (there hemoglobin levels were <11 gm/dL). Out of 387 reviewed chart 344 (88.9%) women have no complication, while 43 (11.1%) have complication like vaginal bleeding 23 (33.3%), severe abdominal pain 18(26.1%), surgical site infection 17 (24.6%) (**See Table 3**).

## 3.4 Factors associated with postpartum anemia after Cesarean delivery

Binary logistic regression analysis was applied to identify factors associated with postpartum anemia after cesarean delivery. In a bivariable logistic regression analysis nine variables such

**Table 2. Obstetrics related characteristics of respondents in public hospitals of Awi zone, Ethiopia, 2023.**

| Variables | Category | Frequency | Percentage (%) |
|---|---|---|---|
| ANC follow up | Yes | 325 | 84 |
| | No | 62 | 16 |
| Gravidity | Primigravida | 105 | 27.1 |
| | Multigravida | 282 | 72.9 |
| Parity | Primipara | 148 | 38.2 |
| | Multipara | 239 | 61.8 |
| Utilization of IFA | Yes | 315 | 81.4 |
| | No | 72 | 18.6 |
| Number of gestations | Single | 356 | 92 |
| | Multiple | 31 | 8 |
| Number of previous C/S | None | 331 | 85.5 |
| | 1 | 32 | 8.3 |
| | >/ = 2 | 24 | 6.2 |
| C/S type | Emergency | 318 | 82.2 |
| | Elective | 69 | 17.8 |
| Indication for current C/S | Fetal distress | 118 | 30.5 |
| | Prior scar | 43 | 11.1 |
| | Prolonged labour | 85 | 22.0 |
| | Multiple pregnancy | 21 | 5.4 |
| | Malpresentation | 50 | 12.9 |
| | preeclampsia-eclampsia | 53 | 13.7 |
| | Other | 17 | 4.4 |
| Type of C/S (based on uterine incision) | LUST C/S | 383 | 99.0 |
| | Classic C/S | 4 | 1.0 |
| Weight of newborn | Average | 360 | 93.0 |
| | Low birth weight | 11 | 2.8 |
| | Macrosomia | 16 | 4.1 |
| Antepartum hemorrhage in the current px | Yes | 43 | 11.1 |
| | No | 344 | 88.9 |
| Post-partum hemorrhage (PPH) | Yes | 28 | 7.2 |
| | No | 359 | 92.8 |
| Post operation Hgb | <11 gm/dl | 73 | 18.9 |
| | >/ = 11 gm/dl | 314 | 81.1 |
| Did transfuse blood | Yes | 14 | 3.6 |
| | No | 373 | 96.4 |
| Any Post operation complication | Yes | 43 | 11.1 |
| | No | 344 | 88.9 |
| If yes which; | Surgical site infection | 17 | 24.6 |
| | Vaginal bleeding | 23 | 33.3 |
| | Severe abdominal pain | 18 | 26.1 |
| | Other | 11 | 15.9 |

as ANC follow up, parity, utilization of IFA, C/S Type, weight of newborn, indication for current C/S, APH, pre-operation hemoglobin and medical complication during current pregnancy were variables that have found to have association with PPA after cesarean delivery at p-value of < 0.25 (See Table 4).

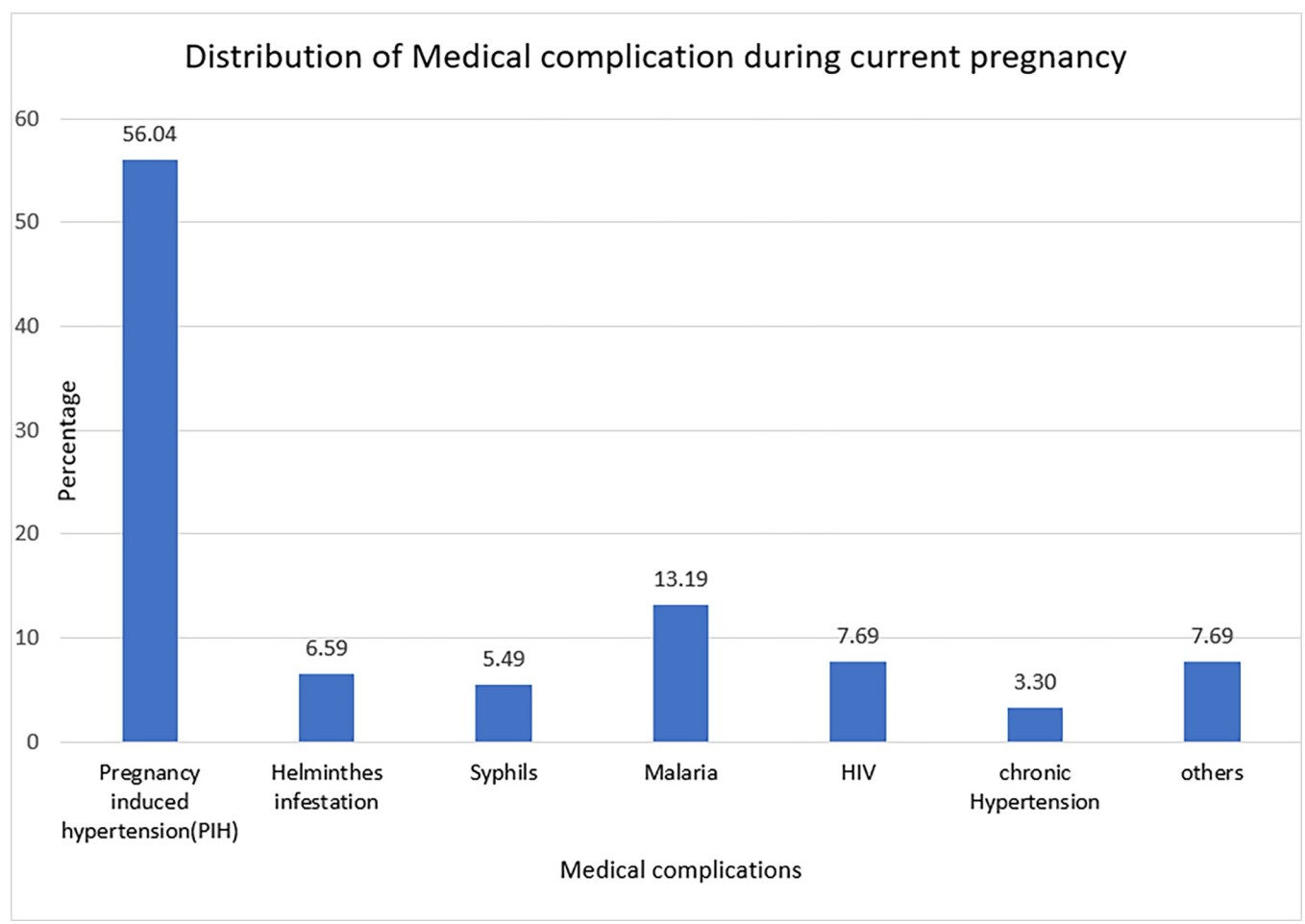

**Fig 2. Distribution of medical complication during current pregnancy among postpartum mother in public hospitals of Awi zone, Ethiopia,2013.**

But only four variables were significantly associated with postpartum anemia after cesarean delivery in multivariable logistic regression such as parity, indication for current C/S, pre-operation hemoglobin and medical complication during current pregnancy (**See Table 4**).

**Table 3. Prevalence and explanatory variables of postpartum anemia after cesarean delivery in public hospitals of Awi zone, Ethiopia 2023.**

| Variable | Categories | Frequency | Percentage (%) |
|---|---|---|---|
| Post operation Hgb | <11 gm/dl | 73 | 18.9 |
| | >/ = 11 gm/dl | 314 | 81.1 |
| Did transfuse blood | Yes | 14 | 3.6 |
| | No | 373 | 96.4 |
| Any Post operation complication | Yes | 43 | 11.1 |
| | No | 344 | 88.9 |
| If yes which; | Surgical site infection | 17 | 24.6 |
| | Vaginal bleeding | 23 | 33.3 |
| | Severe abdominal pain | 18 | 26.1 |
| | Other* | 11 | 15.9 |

Other* = severe headache, nausea and vomiting, DVT, UTI.

**Table 4. Bivariate and multivariable logistic regression analysis, factors associated with postpartum anemia after cesarean delivery in public hospitals of Awi zone, Ethiopia, 2023.**

| Variables | Categories | PPA after C/S | | COR (95% CI) | p-value for bivar. | AOR (95% CI) | p-value |
|---|---|---|---|---|---|---|---|
| | | Yes (%) | No (%) | | | | |
| ANC | Yes | 56(17.2%) | 269(82.8%) | 0.55(0.29,1.03) | 0.063 | 0.99(0.24,4.10) | 0.992 |
| | No | 17(27.4%) | 45(72.6%) | 1 | 1 | 1 | |
| Parity | Primipara | 23(15.5%) | 125(84.5%) | 0.70(0.40,1.20) | 0.190 | 0.47(0.24,0.92) | **0.027** |
| | Multipara | 50(20.9%) | 189(79.1%) | 1 | 1 | 1 | |
| Utilization of IFA | Yes | 54(17.1%) | 261(82.9%) | 0.60(0.32,1.05) | 0.073 | 0.51(0.13,1.93) | 0.320 |
| | No | 19(26.4%) | 53(73.6%) | 1 | 1 | 1 | |
| C/S Type | Emergency | 67(21.1%) | 251(78.9%) | 2.80(1.16,6.75 | 0.022 | 0.43(0.13, 1.49) | 0.185 |
| | Elective | 6(8.7%) | 63(91.3%) | 1 | 1 | 1 | |
| Weight of Newborn | Normal | 65(18.1%) | 295(81.9%) | 1 | 1 | 1 | |
| | LBW | 2(18.2%) | 9(81.8%) | 1.01(0.21,4.78) | 0.991 | 1.45(0.25,8.30) | 0.676 |
| | Macrosomic | 6(37.5%) | 10(62.5%) | 2.72(0.96,7.76) | 0.061 | 2.78(0.85,9.15) | 0.092 |
| Indication for current CS | Fetal distress | 28(23.7%) | 90(76.3%) | 1 | 1 | 1 | |
| | Prolonged labor | 19(22.4%) | 66(77.6%) | 0.92(0.48,1.80) | 0.997 | 1.17(0.55,2.45) | 0.682 |
| | Multiple px | 2(9.5%) | 19(90.5) | 0.34(0.07,1.54) | 0.819 | 0.16(0.02, 1.15) | 0.068 |
| | Malpresentation | 7(14.0%) | 43(86.0%) | 0.52(0.21,1.29) | **0.162** | 0.29(0.09,0.90) | **0.032** |
| | Preeclampsia-eclampsia | 11(20.8%) | 42(79.2%) | 0.84(0.38,1.85 | **0.160** | 0.14(0.03,0.56) | **0.007** |
| | Other | 6(35.3%) | 11(64.7) | 1.75(0.59,5.17) | 0.668 | 1.96(0.59, 6.54) | 0.271 |
| APH | Yes | 12(27.9%) | 31(72.1%) | 1.80(0.87,3.69) | 0.112 | 1.26(0.50,3.18) | 0.628 |
| | No | 61(17.7%) | 283(82.3%) | 1 | 1 | 1 | |
| Pre-operation Hgb | <11gm/dL | 14(70%) | 6(30%) | 12.18(4.50,32.98) | 0.000 | 14.50(4.11,51.16) | **0.000** |
| | >/ = 11gm/dL | 59(16.1%) | 308(83.9%) | 1 | 1 | 1 | |
| Medical complication | Yes | 19(28.4%) | 48(71.6%) | 1.95(1.06,3.58) | 0.031 | 5.95(1.88,18.83) | **0.002** |
| | No | 54(16.9%) | 266(83.1%) | 1 | 1 | 1 | |

The study revealed that the odds of postpartum anemia after cesarean delivery were 53% lower among mothers who were primipara compared to those who were multipara (AOR = 0.47, 95% CI = 0.24,0.92).

In contrast to those mothers whose indication for current C/S was fetal distress, the odds of postpartum anemia after cesarean delivery were 71% and 86% times lower among mother whose indication for current C/S were malpresentation and preeclampsia-eclampsia, respectively, when compared with those mothers whose indication for current C/S were fetal distress (AOR = 0.29, 95% CI = 0.09,0.90) (AOR = 0.14, 95% CI = 0.03,0.56).

In addition, this study revealed that postpartum anemia after CD was strongly associated with a predelivery Hb level; the odds of postpartum anemia after CD were 14.5 times higher among women with predelivery anemia (Hb level < 11gm/dL), compared to those who had normal predelivery Hb level (>/ = 11gm/dL) (AOR = 14.5; 95% CI = 4.11, 51.16).

Furthermore, having medical complication during the last pregnancy was also significantly associated with of postpartum anemia after cesarean delivery, in which the odds of postpartum anemia after cesarean delivery were 5.95 times more likely among postpartum mother with clinically confirmed medical complication as compared to their counterparty (AOR = 5.95, 95% CI = 1.88, 18.83).

## 4. Discussion

This study attempted to assess the prevalence and associated factors of postpartum anemia after cesarean delivery in public hospitals of Awi Zone. The study revealed that the prevalence of postpartum anemia after cesarean delivery was 18.9% (95% CI (15.1, 23.1)).

This finding was in line with the study done in Nigeria which reported that prevalence of postpartum anemia after cesarean delivery was 20.8% [30], and in Debreberhan (18%) [31] and in Debre Markos (22.87%) [32]. However, the prevalence of PPA after cesarean delivery in this study was higher than other studies conducted in Karimnagar of India (9.2%) [33], This discrepancy might be due to study settings difference, as the above studies in Debreberhan and Debre Markos were conducted at tertiary hospitals. Although, in this study lowest threshold used was less than 11g/dl hemoglobin level.

On the other hand, this finding was lower than those of studies conducted in Turkey (50.85%) [34], Mumbai(76.5%) [35], Kenya (25%) [9] and Harari of Ethiopia (47.4%) [36]. This difference might be due to the difference in severely anemic mothers in prior to operation being excluded from this study, use of different cutoff points to define postpartum anemia, and difference in postpartum time of screening. Due to a lack of consensus on the definition of PPA, different scholars use different cutoff points to diagnose PPA. In addition. Geographical difference might be also another factor for the abovementioned variation. Additionally, Ethiopians are eating foods that do have high iron content such as cereals, "teff injera"(ferment teff flour), and fruits [37].

The study revealed that the odds of postpartum anemia after cesarean delivery were lower among mothers who were primipara compared to those who were multipara. This finding is consistent with a cross-sectional study conducted in south India which showed that parity of two or more was significantly associated with postpartum anemia [38]. This finding is also supported by Cross-sectional study conducted in at Debre Berhan referral hospital of Ethiopia [31]. The reason might be due to in case of multiparity the muscular strength of the uterus reduced due to the loss of collagen fibers, results decreased uterine contraction after birth leads to blood loss.

In this study, the odds of postpartum anemia after cesarean delivery were lower among mother whose indication were malpresentation and preeclampsia-eclampsia, respectively, than those mothers whose indication were fetal distress, The reason may be due to in case of malpresentation and preeclampsia-eclampsia health care providers consciously follow laboring mothers take immediate action soon after the diagnosis due to fear of complication this may reduce predelivery blood loss. The other possible explanation might be due to in the case of fetal distress emergency C/S is mostly performed type of C/S which results more blood loss. According to studies, postoperative complications were found higher in emergency Cesarean section as compared to elective Cesarean section like postpartum anemia (70% vs. 40%) and postpartum hemorrhage (40% vs. 6%) [39].

In addition, this study revealed that postpartum anemia after CD was strongly associated with a predelivery Hb level; the odds of postpartum anemia after CD were 14.5 times higher among women with predelivery Hb level of less than 11gm/dL, compared to those who had normal predelivery Hb level (>/ = 11gm/dL). This finding is consistent with studies conducted in California and India [29,40]. It is also supported by studies conducted in Uganda, Nigeria and Harari of Ethiopia [28,30,36]. The possible explanation for this might be because of the women who had preoperative anemia were less tolerant of any amount of blood loss during cesarean section. The possible explanation for this might be because the iron deficiency present during the antenatal period continues through the postpartum period also.

In this study, having medical complication during the last pregnancy was also significantly associated with of postpartum anemia after cesarean delivery. The odds of postpartum anemia after cesarean delivery were more likely among postpartum mother with clinically confirmed medical complication. This finding is consistent with a cross sectional study done in Nairobi of Kenya [9] which showed that complications during pregnancy was significantly associated with postpartum anemia after cesarean delivery. This finding is also supported by study

conducted in Nigeria [30] which reported that co-morbidities like hypertensive disorders in pregnancy was statistically associated with postpartum anemia after cesarean delivery.

### 4.1 Limitation of the study

As the study relies on a retrospective review of data, some important variables that can help to generalize the findings were lack.

## 5. Conclusion

This study indicated that the prevalence of PPA after cesarean delivery is 18.9%. Which categorized under a mild public health problem per the WHO cut-off value for the public health significance of anemia. Parity, indication for current C/S, pre-operation hemoglobin level($<$11g/dl) and having medical complication during current pregnancy were factors significantly associated with postpartum anemia after cesarean delivery.

## 6. Recommendation

Determining patients under high risk (having predelivery anemia and medical complication during pregnancy) is still important to be alert to reduce postpartum anemia after cesarean delivery. Prevention, early detection and treatment of predelivery anemia, could reduce postpartum anemia after cesarean delivery. Also, health care providers need to consciously follow laboring mothers with medical complications. In addition, management of anemia at the antenatal period is the most crucial strategy in combating PPA after CD and this is highly recommended in all levels of health care system through FMOH. Furthermore, Further research better to be conducted that can address the limitations of this study and design strategies to improve completeness by using prospective study design.

## Supporting information

**S1 File. SPSS dataset.**
(SAV)

## Acknowledgments

The authors thank the data collectors, data collectors' supervisors, and administrative staff of all hospitals for their cooperation to conduct this research.

## Author Contributions

**Conceptualization:** Gebretsadek Habtamu, Asmare Talie, Dawit Misganaw Belay.

**Data curation:** Gebretsadek Habtamu, Asmare Talie, Tinsae Kassa.

**Formal analysis:** Gebretsadek Habtamu, Dawit Misganaw Belay.

**Funding acquisition:** Gebretsadek Habtamu.

**Investigation:** Gebretsadek Habtamu, Asmare Talie, Tinsae Kassa, Dawit Misganaw Belay.

**Methodology:** Gebretsadek Habtamu, Asmare Talie, Tinsae Kassa, Dawit Misganaw Belay.

**Resources:** Gebretsadek Habtamu.

**Software:** Gebretsadek Habtamu, Dawit Misganaw Belay.

**Supervision:** Asmare Talie, Tinsae Kassa.

**Validation:** Asmare Talie, Tinsae Kassa, Dawit Misganaw Belay.

**Visualization:** Gebretsadek Habtamu, Asmare Talie, Tinsae Kassa, Dawit Misganaw Belay.

**Writing – original draft:** Gebretsadek Habtamu, Dawit Misganaw Belay.

**Writing – review & editing:** Gebretsadek Habtamu, Asmare Talie, Tinsae Kassa, Dawit Misganaw Belay.

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
