## [Decision Letter · Decision Letter 0]

9 Jun 2024

PONE-D-24-07591Magnitude and Associated Factors of Postpartum Anemia After Caesarean Delivery in Public Hospitals of Awi Zone, North West Ethiopia, 2023; A Cross-Sectional Study.PLOS ONE

Dear Dr. Belay,

Thank you for submitting your manuscript to PLOS ONE. After careful consideration, we feel that it has merit but does not fully meet PLOS ONE’s publication criteria as it currently stands. Therefore, we invite you to submit a revised version of the manuscript that addresses the points raised during the review process.

**ACADEMIC EDITOR: **

**Please respond to all reviewers comments point by point **

We look forward to receiving your revised manuscript.

Kind regards,

Ahmed Mohamed Maged, MD

Academic Editor

PLOS ONE

2. Please ensure the data collection period is accurate in the manuscript.

3. In this instance it seems there may be acceptable restrictions in place that prevent the public sharing of your minimal data. However, in line with our goal of ensuring long-term data availability to all interested researchers, PLOS’ Data Policy states that authors cannot be the sole named individuals responsible for ensuring data access (http://journals.plos.org/plosone/s/data-availability#loc-acceptable-data-sharing-methods).

Reviewers' comments:

Reviewer's Responses to Questions

**Comments to the Author**

1. Is the manuscript technically sound, and do the data support the conclusions?

Reviewer #1: Yes

Reviewer #2: No

Reviewer #3: Partly

Reviewer #4: Partly

2. Has the statistical analysis been performed appropriately and rigorously? 

Reviewer #1: Yes

Reviewer #2: No

Reviewer #3: Yes

Reviewer #4: I Don't Know

3. Have the authors made all data underlying the findings in their manuscript fully available?

Reviewer #1: Yes

Reviewer #2: No

Reviewer #3: Yes

Reviewer #4: Yes

4. Is the manuscript presented in an intelligible fashion and written in standard English?

Reviewer #1: No

Reviewer #2: No

Reviewer #3: No

Reviewer #4: No

5. Review Comments to the Author

Reviewer #1: 1. Do you have full confidence to say this study, a cross sectional?

2. What was the basic gap for this study? What if there is no study in study area and little is known about it? you are well expected to show the impact of little knowledge about PPA after CD?

3. Why do you say this study is cross-sectional? Since, it used The card numbers of all mothers who gave birth through cesarean section at Awi Zone public hospital for the last two years before the study (January 1, 2021 to December 31, 2022) were traced from the hospital’s delivery log book registry and were listed down. A two-year report of birth through cesarean section is collected from each hospital health management information system and sum up to calculate proportion?

4. Have you ever tried the reverse of these relations to explore positive association? If, yes why not presented in result section?

5. Sections from study design to eligiblity criteria could be deduced in a single paragraph/sentence.

6.Double population proportion should be used for two independent populations and used for study designs compare two different populations. So, why you compute DPPF for this cross-sectional study?

7. please, be consistent with your study objective/title and/operational definition (for surgical site infection 17 (24.6%)). Or again operationally define this phrase.

8. Indication for current CS was not significant in bivariate. So, why you used in multivariate analysis?

9. revise the conclusion section because, a conclusion of a manuscript should be very precise, short and comprehensive.

10. limitation lacks not only treatment outcome, which is not the concern of this study. It lack variables that can help to generalize the findings.

11. in General, the manuscript needs critical revision including grammatical correction and keeping the standard based on instruction for authors

Reviewer #2: 1- Title can be simplified as prevalence of post partum anemia :An observational study

2- Abstract contains unknown abbreviation eg AOR?

3- The major defect in this cross sectional is lacking control group? As it must be compared with non anemic patients so you must compare anemic patients versus non anemic patients as regard age ,parity, previous morbidity and so on

4- So this is not cross sectional study please compare your data to control group otherwise it lacks credibility or significance

Reviewer #3: Conclusion:

The findings of the study show that the magnitude of postpartum anemia after cesarean delivery is a mild public health problem. Therefore, promoting the benefits of early detection and management of pregnancy complications such as predelivery anemia and medical complications is crucial.

Comment : the conclusion does not refer to or clarify to the paper title i.e. no comment about associated factors

Sampling procedure :

Comment : concerning cases who had postpartum hemorrhage or blood transfusion , when was the HB sample collected ? before or after the hemorrhage or before or after transfusion ?

Obstetrics related characteristics of mother:

The most common type of CS was emergency 318 (82.2%)

Comment : very high percentage of emergency Cs , what is the reason behind that ? and as mentioned in table 3 : 30 % was due to fetal distress which is also a very high percentage

Table 3 :

Comment : Type of pregnancy : should be renamed to number of gestations : singleton or multiple

Weight of newborn

Comment : change normal to average

Comment : change Gravida to Gravidity

Study Area and Period

Comment ; you should mention the number of deliveries annually in the area

Table 4

Comment : why include severe abdominal pain as a complication ? and what was the cause of the pain ?

CONCLUSION AND RECOMMENDATION

Comment :You should mention that you should reduce the emergency Cs rates as it is related to anemia and elaborate a bit about how you can do that in your setting

English

Comment : the English should be revised before publishing

Reviewer #4: Dear Editor, thank you much for inviting me to review this manuscript. I forwarded the following comments and recommendations to the authors.

Title: Replace the term “Magnitude” with prevalence.

Introduction:

Line 40-42: “It can be defined as Hgb < 10 g/dl, Hgb < 11g/dl, and Hgb < 12g/dl cut off values within the first 48 h of delivery, at 1 week and 6 weeks of postpartum duration, respectively(2, 3). Since you put different three cut-off value, you would cite all the sources. Again write ‘hour’ in full.

Line 72: “Additionally, … “ Avoid this conjunction because the earlier paragraph states about attributes of postpartum anemia.

“Despite positive progress being made in many countries to reduce maternal mortality, there is still evidence of a persistent increase in the rate of indirect causes” cite the sources.

“Several studies have been conducted 78 on anemia during pregnancy” cite the sources

Materials and Methods:

The authors would gave more detail the study settings (hospitals), including the types of maternal services, estimated number of population being served, number skilled personnel… etc.

How the authors could include all two years cesarean deliveries the settings? Was it census study?

The authors excluded postpartum women who had pre-operative severe anaemia. But they did not define severe anemia either in introduction or methods sections.

Line 123 All the public hospitals in the Zone were included.

Line 13: “The extent of postpartum anemia after caesarean delivery as a dependent variable, defined by World Health Organization (WHO) criteria as a postpartum Hgb level of less than 11 g/dL, measured closest to the day of hospital discharge (27).” That means you have missed possible causes of anemia that could occur after discharge

Line 144: Postpartum anemia after Caesarean delivery what; incidence, prevalence, or what? Please, make it clear and understandable for readers.

Each independent variable would be depicted.

Line 176 & 177: “The final model fitness was checked using the Hosmer-Lemeshow Goodness of Fit test 177 and multicollinearity test were done to check the relationship between independent variables.” Revise this sentence.

Results:

Line 188 &189: “There were 1995 deliveries in 2021 and 2688 deliveries in 2022, were delivered by CS in five public hospitals of Awi zone.” a confusing statement.

The study missed important sociodemographic and dietary or nutritional-related attributes of anemia .

I suggest he authors to include “Medical conditions related characteristics” in the “Obstetrics related characteristics of mother” because they did not mention any non-medical condition under this section. Two mentioned conditions (anemia during pregnancy and PIH) are obstetric related characteristics.

The authors would distinguish the degrees of anemia (depending on the severity) in their study that is very important management options and prioritizations.

The authors would include the p-values of variables associated in bivariable analysis.

The authors did not described “medical related conditions in the last pregnancy” as a variable in the descriptive statistics but they analysed it as independent variable predictor of postpartum anemia.

Discussion:

You would not compare your finding with the findings from California and Uganda those used Hgb level <8 g/dl and <7g/dl, respectively, a cut-off value.

The authors compared and contracted their proportions with incidence rates. For instance reference number 33 states about incidence rate postpartum anemia.

The authors would not compare their finding with the proportion of anemia among lactating mothers in subsistence farming households. Different population characteristics and diffirent settings. Reference number 35.

Line 275-277: Would be included in first paragraph or omitted.

General comments:

This manuscript has major tense, grammar, punctuation, and sentence errors. So the authors must get help from professional language editors.

6. PLOS authors have the option to publish the peer review history of their article (what does this mean?). If published, this will include your full peer review and any attached files.

Reviewer #1: No

Reviewer #2: **Yes: **Adel Mohamed Nada

Reviewer #3: **Yes: **Hassan Gaafar

Reviewer #4: No

---

## [Author Response · Author response to Decision Letter 0]

24 Jun 2024

“Prevalence and Associated Factors of Postpartum Anemia After Caesarean Delivery in Public Hospitals of Awi Zone, North West Ethiopia, 2023; A Cross-Sectional Study."

Dear Editors and reviewers 

Thank you for sending us your valuable comments, which immensely improved our manuscript. We included all the editorial comments raised and enclosed the point-by-point response attached. It is my pleasure to inform you that the manuscript was edited meticulously.

Respectfully, 

Dawit Misganaw Belay

Corresponding author

---

## [Decision Letter · Decision Letter 1]

29 Jul 2024

PONE-D-24-07591R1Prevalence and Associated Factors of Postpartum Anemia After Caesarean Delivery in Public Hospitals of Awi Zone, North West Ethiopia, 2023; A Cross-Sectional Study.PLOS ONE

Dear Dr. Belay,

Thank you for submitting your manuscript to PLOS ONE. After careful consideration, we feel that it has merit but does not fully meet PLOS ONE’s publication criteria as it currently stands. Therefore, we invite you to submit a revised version of the manuscript that addresses the points raised during the review process.

**ACADEMIC EDITOR: Please respond to all reviewers comments**

We look forward to receiving your revised manuscript.

Kind regards,

Ahmed Mohamed Maged, MD

Academic Editor

PLOS ONE

Reviewers' comments:

Reviewer's Responses to Questions

**Comments to the Author**

1. If the authors have adequately addressed your comments raised in a previous round of review and you feel that this manuscript is now acceptable for publication, you may indicate that here to bypass the “Comments to the Author” section, enter your conflict of interest statement in the “Confidential to Editor” section, and submit your "Accept" recommendation.

Reviewer #1: (No Response)

Reviewer #3: All comments have been addressed

2. Is the manuscript technically sound, and do the data support the conclusions?

Reviewer #1: Yes

Reviewer #3: Yes

3. Has the statistical analysis been performed appropriately and rigorously? 

Reviewer #1: No

Reviewer #3: Yes

4. Have the authors made all data underlying the findings in their manuscript fully available?

Reviewer #1: Yes

Reviewer #3: Yes

5. Is the manuscript presented in an intelligible fashion and written in standard English?

Reviewer #1: Yes

Reviewer #3: Yes

6. Review Comments to the Author

Reviewer #1: mOST OF THE COMMENTS I FORWADED HAVE NOT BEEN COMPLETE. tHUS, THE AOUTHORS ARE EXPECTED TO DO SO BEFORE PUBLICATION OF THE MANUSCRIPT.

Reviewer #3: thank you for replying to all my comments

i think the paper is now ready to be published

next time try to use better English and address all points of the title of the paper in the discussion

7. PLOS authors have the option to publish the peer review history of their article (what does this mean?). If published, this will include your full peer review and any attached files.

Reviewer #1: No

Reviewer #3: **Yes: **Hassan Gaafar

---

## [Author Response · Author response to Decision Letter 1]

5 Aug 2024

Dear Editors and reviewer 

Thank you for sending us your valuable comments, which have immensely improved our manuscript. We have included all the editorial comments raised and attached a point-by-point response. 

However, Reviewer 1 did not see our previously revised version of the manuscript and still commented on the first submitted manuscript. It is my pleasure to inform you that the manuscript has been edited meticulously.

RESPONSE FOR REVIEWER #1: 

Reviewer #1: MOST OF THE COMMENTS I FORWADED HAVE NOT BEEN COMPLETE. THUS, THE AOUTHORS ARE EXPECTED TO DO SO BEFORE PUBLICATION OF THE MANUSCRIPT.

Response: Thank you, dear reviewer, for your valuable feedback. However, most of your current comments were addressed in the revised version of our manuscript, but it seems you are still commenting on the first submitted manuscript, which is why you may still see the same gaps. Please refer to the revised version of our manuscript and the attached response to reviewers. Additionally, some new comments have been incorporated into this revised manuscript. 

In addition, comment you raised regarding the figure caption, the manuscript is prepared based on the “PLOS ONE Figure Guideline”; which said “Do not include figures in the main manuscript file. Each figure must be prepared and submitted as an individual file. Figure captions must be inserted in the text of the manuscript, immediately following the paragraph in which the figure is first cited (read order). Do not include captions as part of the figure files themselves or submit them in a separate document.”

Thank you once again for your valuable comments and suggestions.

RESPONSE FOR REVIEWER #3: 

1. thank you for replying to all my comments. I think the paper is now ready to be published

next time try to use better English and address all points of the title of the paper in the discussion.

Response: thank you for your scholarly comments and suggestions.

---

## [Editor Report · Decision Letter 2]

27 Sep 2024

Prevalence and Associated Factors of Postpartum Anemia After Caesarean Delivery in Public Hospitals of Awi Zone, North West Ethiopia, 2023; A Cross-Sectional Study.

PONE-D-24-07591R2

Dear Dr. Belay,

We’re pleased to inform you that your manuscript has been judged scientifically suitable for publication and will be formally accepted for publication once it meets all outstanding technical requirements.

Kind regards,

Ahmed Mohamed Maged, MD

Academic Editor

PLOS ONE
---

## [Editor Report · Acceptance letter]

5 Nov 2024

PONE-D-24-07591R2 

PLOS ONE

Dear Dr. Belay, 

I'm pleased to inform you that your manuscript has been deemed suitable for publication in PLOS ONE. Congratulations! Your manuscript is now being handed over to our production team.

Kind regards, 

on behalf of

Professor Ahmed Mohamed Maged 

Academic Editor

PLOS ONE